# H$_2$O$_2$ and Engrailed 2 paracrine activity synergize to shape the zebrafish optic tectum

Irène Amblard[1,2,8], Marion Thauvin[1,2,8], Christine Rampon[1,3], Isabelle Queguiner[1], Valeriy V. Pak[4,5], Vsevolod Belousov[4,5,6], Alain Prochiantz[1], Michel Volovitch[1,7], Alain Joliot [1,9✉] & Sophie Vriz [1,3,9✉]

Although a physiological role for redox signaling is now clearly established, the processes sensitive to redox signaling remains to be identified. Ratiometric probes selective for H$_2$O$_2$ have revealed its complex spatiotemporal dynamics during neural development and adult regeneration and perturbations of H$_2$O$_2$ levels disturb cell plasticity and morphogenesis. Here we ask whether endogenous H$_2$O$_2$ could participate in the patterning of the embryo. We find that perturbations of endogenous H$_2$O$_2$ levels impact on the distribution of the Engrailed homeoprotein, a strong determinant of midbrain patterning. Engrailed 2 is secreted from cells with high H$_2$O$_2$ levels and taken up by cells with low H$_2$O$_2$ levels where it leads to increased H$_2$O$_2$ production, steering the directional spread of the Engrailed gradient. These results illustrate the interplay between protein signaling pathways and metabolic processes during morphogenetic events.

[1] Center for Interdisciplinary Research in Biology (CIRB), Collège de France, CNRS, INSERM, PSL Research University, Paris, France. [2] Sorbonne Université, Paris, France. [3] Université de Paris, Faculty of Sciences, Paris, France. [4] Center for Precision Genome Editing and Genetic Technologies for Biomedicine, Pirogov Russian National Research Medical University, Moscow 117997, Russia. [5] Institute for Cardiovascular Physiology, Georg August University Göttingen, 37073 Göttingen, Germany. [6] Federal Center of Brain Research and Neurotechnologies, FMBA, Moscow 117997, Russia. [7] Department of Biology, École Normale Supérieure, PSL Research University, Paris, France. [8] These authors contributed equally: Irène Amblard, Marion Thauvin. [9] These authors jointly supervised this work: Alain Joliot, Sophie Vriz. ✉email: alain.joliot@college-de-france.fr; vriz@univ-paris-diderot.fr

Reactive oxygen species (ROS), including hydrogen peroxide ($H_2O_2$), once only considered as deleterious compounds, have recently raised novel interest due to their action as bona fide-signaling molecules[1–3]. Ratiometric probes selective for $H_2O_2$ have revealed its complex spatiotemporal dynamics during neural development and adult regeneration[4,5]. In addition, modifying $H_2O_2$ levels disturbs cell plasticity and morphogenesis[6,7]. Proteins targeted by $H_2O_2$ during development, by mechanisms still largely elusive, belong to many categories and include homeoproteins (HPs)[8,9]. HPs play important roles in the control of cellular and regional identity during development. Although first characterized as purely cell autonomous transcription factors[10], HPs are also transferred between cells via non-conventional secretion and internalization routes, providing them with direct paracrine activity[11]. Intercellular transfer is a general property of HPs, and its efficiency is context-dependent[12,13]. Within recipient cells, transferred HPs may act not only as transcription factors but also on processes as diverse as translation, DNA repair, mitochondrial activity, and epigenetic modification[14].

We have previously shown that the zebrafish midbrain–hindbrain boundary (MHB) displays high levels of $H_2O_2$ that cannot be lowered without affecting tectum topography[6], a process that also requires the graded distribution of the Engrailed HPs (EN in amniotes, Eng in fish)[15–18]. We thus asked whether $H_2O_2$ levels and Engrailed distribution interplay with each other by conducting in parallel a close examination of both parameters at the time of the tectum anteroposterior polarization. Importantly, the Engrailed extracellular gradient is instrumental in establishing tectum polarity in frogs and chicks[19], requires Eng2b paracrine activity in the zebrafish[20]. We combined ex vivo and in vivo approaches to demonstrate an unsuspected role of $H_2O_2$ in Engrailed homeoprotein spreading during tectum development. Engrailed is released from cells with high $H_2O_2$ levels and transfer to cells with low $H_2O_2$ levels in which it stimulates $H_2O_2$ production, thereby controlling its own polarized traffic. In addition, we identify cysteine 175 as a key residue in the redox regulation of Engrailed traffic.

## Results and discussion

### $H_2O_2$ levels shape Engrailed 2 distribution in the tectum.

The graded distribution of Eng2 evolves during development, particularly between 24 and 26 h post fertilization (hpf) (Fig. 1a, b and Supplementary Fig. 1). As previously shown in several vertebrates[15–18], Eng2 level is maximum at the MHB where it is known to be synthesized and decreases along the posterior–anterior axis until becoming undetectable at the most anterior part of the tectum. The temporal analysis of Eng2 distribution throughout the tectum (24–28 hpf) showed that, at the MHB, Eng2 level is highest at 24 hpf and decreased at 26 and 28 hpf (Fig. 1b). These modifications in Eng2 distribution were concomitant with remarkable changes in $H_2O_2$ levels in the same structure (Fig. 1d, e). In the whole tectum, $H_2O_2$ levels increased over time (Fig. 1e), but with a marked gradient from the MHB to the most anterior part of the tectum (Fig. 1d). We noticed that in zebrafish embryos, the clear graded distribution of nuclear Eng2 proteins is accompanied by an extranuclear gradient in the tectum along the anteroposterior axis (Fig. 1a′–a′′′), previously observed in chick and Xenopus embryos[19] and shown to correlate with Engrailed intercellular transfer ex vivo[21]. Using DAPI staining as a marker of the nuclear compartment, nuclear and extranuclear Eng2 signals were quantified separately. The ratio of nuclear to extranuclear Eng2 staining showed that the cellular distribution of Eng2 varied both spatially and temporally along the antero-posterior axis of the tectum from 24 to 28 hpf (Fig. 1c).

The nuclear/extranuclear ratio was highest at 26 hpf and decreased at 28 hpf while $H_2O_2$ increased between 24 and 28 hpf. We thus concluded that EN2 distribution is regulated, at least partially, by $H_2O_2$ levels between 24 and 28 hpf, a temporal window during which both are highly dynamic.

We next tested whether altering endogenous $H_2O_2$ levels would affect Eng2 distribution. Thanks to the use of the improved ratiometric $H_2O_2$ sensor HyPer7[22], we were able to detect, in vivo, the modulation of $H_2O_2$ levels after treatment with a pan-NADPH oxidase inhibitor (Nox-i) (Fig. 1f). The resulting mild decrease in $H_2O_2$ levels led to a marked modification of Eng2 distribution, enhancing Eng2 signal at the MHB but raising the gradient slope in the tectum (Fig. 1g). The higher amount of Eng2 at the MHB was not a consequence of a higher transcription rate since a quantitative RT-PCR performed on the two sets of embryos (Nox-i treated and control embryos) revealed no difference in the amount of eng2a and eng2b mRNA (Supplementary Fig. 2). This suggested that lowering endogenous $H_2O_2$ levels reduced Eng2 spreading from its area of production, most likely by modifying its intercellular trafficking. To test this hypothesis, we analyzed the nuclear/extranuclear distribution of Eng2 over the tectum in embryos treated or not with Nox-i (Fig. 1h). In Nox-i-treated embryos, the cellular distribution of Eng2 was significantly altered (Fig. 1h), characterized by a specific increase of Eng2 in the nuclear fraction, indicating that the correct propagation of Eng2 is dependent on $H_2O_2$ having reached a given threshold. In summary, mild reduction in $H_2O_2$ levels induced both a strong modification in the allocation of Eng2 between the nuclear and extranuclear compartments, and a distorted Eng2 distribution over the tectum. These results suggested that Eng2 intercellular transfer is regulated by $H_2O_2$. To directly address the involvement of $H_2O_2$ in Engrailed intercellular trafficking, we used ex vivo cell culture models which permit an accurate quantification of this process.

### Engrailed 2 intercellular transfer is asymmetrically regulated by $H_2O_2$.

To evaluate the sensitivity of Engrailed trafficking to $H_2O_2$ levels, we performed assays in HeLa cells, where this process has been best charaterized[13,21,23]. $H_2O_2$ levels were fine-tuned with two strategies. To enhance $H_2O_2$ levels, we added D-Alanine (D-Ala) to cells expressing a membrane-bound form of D-amino-acid oxidase (Lck-DAO) (Supplementary Fig. S3a, b)[24]. To reduce $H_2O_2$ levels, we expressed Catalase (CAT) deprived of its peroxisome targeting sequence ($CAT_{\Delta C}$) and targeted to the plasma membrane (Lck-$CAT_{\Delta C}$, Supplementary Fig. S3c, d). We also reduced $H_2O_2$ levels by adding purified CAT to cell culture or by treating cells with Nox-i (Supplementary Fig. S3e, f). Under all conditions, $H_2O_2$ levels were monitored with HyPer expressed in the cytoplasm[25]. The two steps of intercellular transfer, secretion, and internalization, were analyzed separately using dedicated assays set up with the Engrailed 2 homeoprotein (EN2).

To study EN2 secretion, this process was quantified with a new strategy (transRUSH[26]) adapted from the RUSH system[27] (Fig. 2a). Two tags were added to EN2: one (SBP-tag) that hooks the protein at the inner side of the plasma membrane when co-expressed with a membrane-bound Streptavidin hook, and another tag (HiBiT), a small nanoluciferase fragment that allows light production[28] upon interaction with its large counterpart fragment (LgBit) addressed to the extracellular side of the plasma membrane. In this system, intracellular-trapped EN2 is released upon biotin addition, and secretion is monitored by light production upon interaction of EN2-HiBiT with LgBiT at the cell surface. Increasing $H_2O_2$ levels

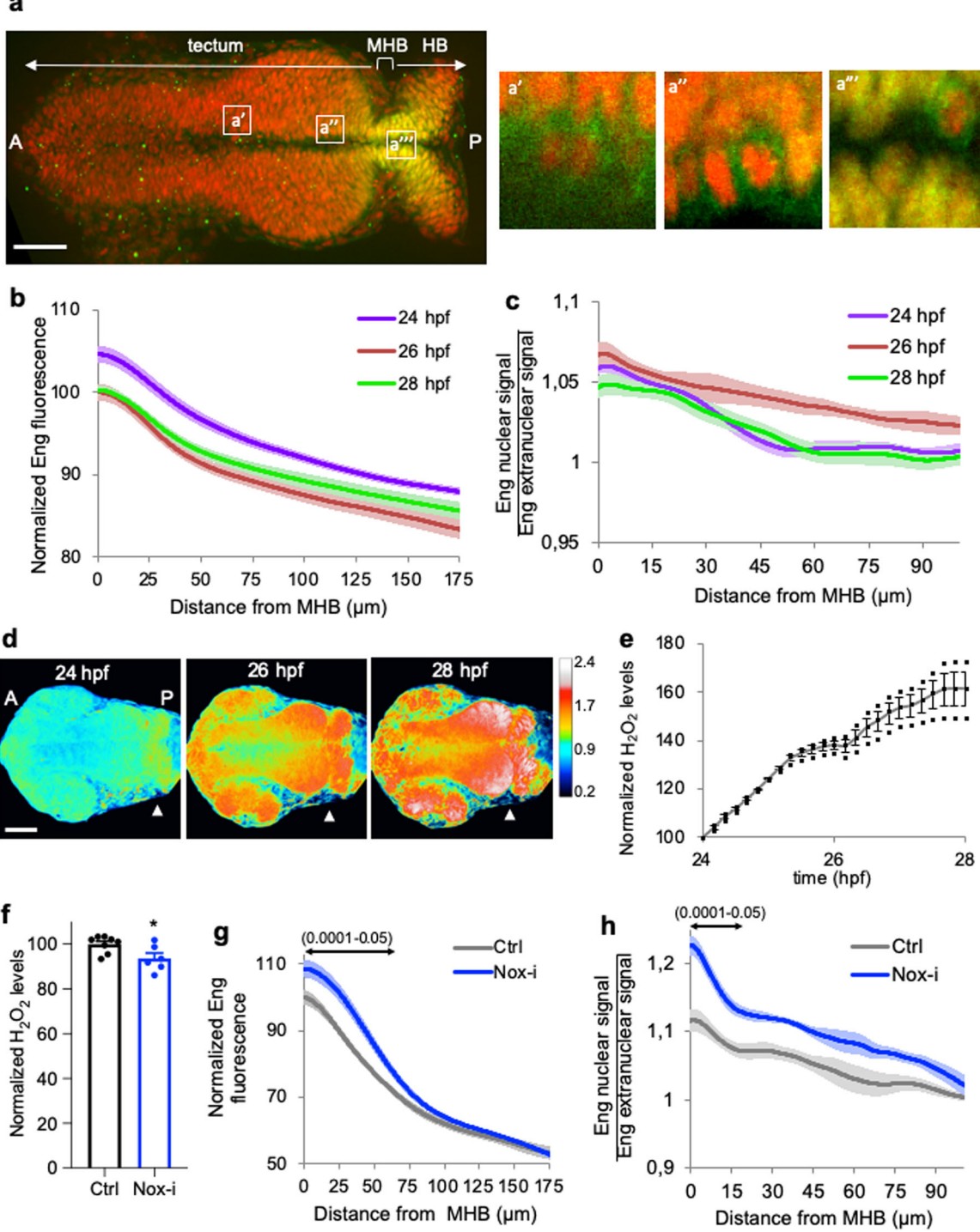

**Fig. 1 $H_2O_2$ levels shape the Engrailed 2 distribution in the tectum. a** Immunodetection of Eng2a and Eng2b (green) and DAPI staining (red) in zebrafish embryos (24 hpf) revealed different nuclear/extranuclear distributions along the anteroposterior axis (a′–a‴: insets of sections at higher magnification; MHB midbrain hindbrain boundary, HB hindbrain). Eng2 staining alone is shown in Supplementary Fig. 1. **b** Quantification of total Eng2 levels (inferred from Eng immunostaining) along the anteroposterior axis of the tectum at 24, 26, and 28 hpf. All values were normalized to the maximum value for 26 hpf. **c** Ratio of Eng2 nuclear over extranuclear signals at 24, 26, and 28 hpf. **d** $H_2O_2$ levels in the tecta of zebrafish embryos from 24 to 28 hpf. $H_2O_2$ levels were inferred from the $YFP_{500}/YFP_{420}$ excitation ratio of HyPer7 in time-lapse recordings. Arrowhead: MHB position. The quantification is presented in **e**. **f** $H_2O_2$ levels in the tecta of control (Ctrl) and two hours Nox-i (100 nM) treated zebrafish 26 hpf embryos. **g** Quantification of immunodetected Eng2 along the anteroposterior axis of the tectum in control (Ctrl) and Nox-i-treated embryos (26 hpf). **h** ratio of Eng2 nuclear/extranuclear signals in control (Ctrl) and Nox-i-treated embryos (26 hpf). Double arrows in **g** and **h** indicate the domain where nuclear to extranuclear ratios are statistically different (confidence interval varying from 0.0001 to 0.05 from the MHB to more anterior positions). Scale bars, 50 μm. *$p$-value ≤ 0.05.

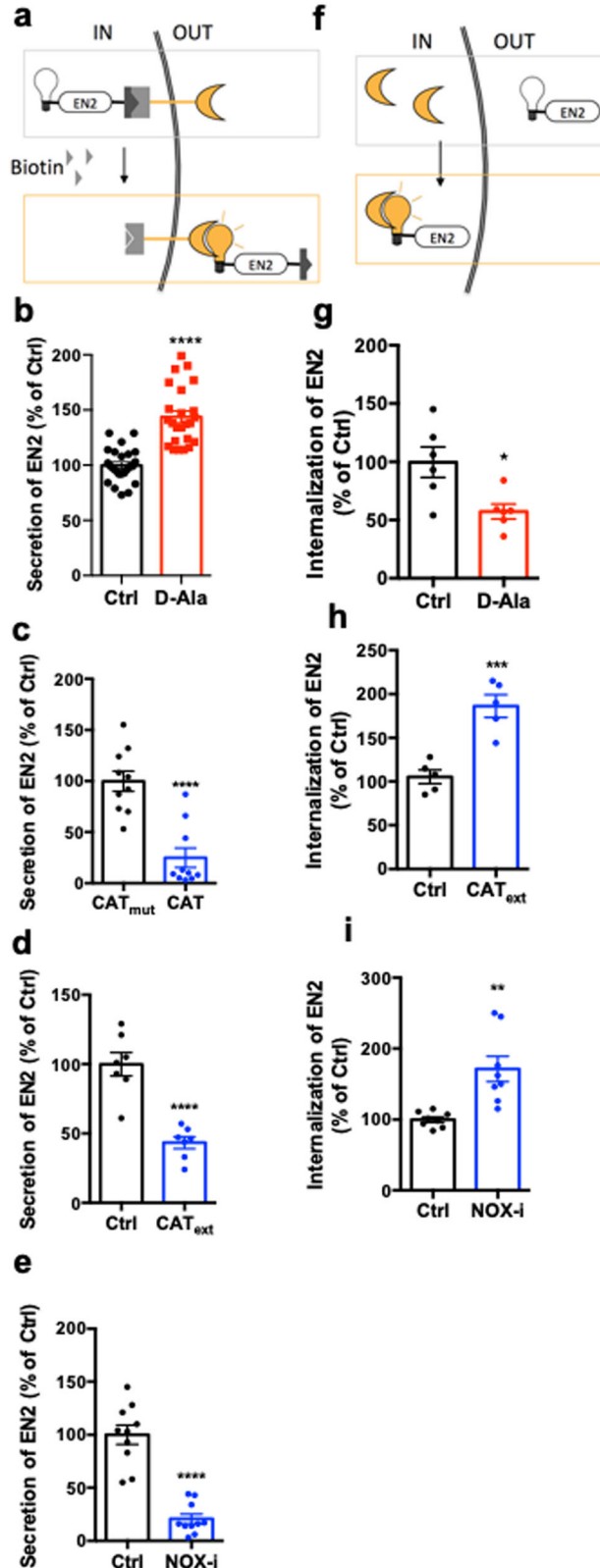

**Fig. 2 Engrailed 2 intercellular transfer is asymmetrically regulated by $H_2O_2$ in HeLa cells. a** Method for quantification of EN2 secretion synchronized at the plasma membrane via a modified RUSH strategy, the transRUSH method[26]. EN2 is tagged with SBP (black) that hooks the protein at the inner side of the plasma membrane in presence of a Streptavidin hook (gray) and HiBiT (white bulb) that emits light when combined to LgBit (yellow crescent). **b–e** Quantification of EN2 secretion from cells expressing Lck-DAO with or without D-Ala **b**, expressing inactive or active Catalase ($CAT_{mut}$ or CAT, respectively) **c**, treated with extracellular CAT ($CAT_{ext}$) **d**, or treated with Nox-i **e**. **f** Method for quantification of EN2 internalization. **g–i** Quantification of EN2 internalization in cells expressing Lck-DAO with or without D-Ala **g**, treated with $CAT_{ext}$ **h** or Nox-i **i**. *$p$-value ≤ 0.05; **$p$-value ≤ 0.01; ***$p$-value ≤ 0.001; and ****$p$-value ≤ 0.0001.

from its interaction with a cytosolic LgBiT (Fig. 2f). Increasing $H_2O_2$ levels (via D-Ala addition) reduced EN2 internalization (Fig. 2g), while decreasing them (by treatment with either purified CAT or Nox-i) enhanced EN2 internalization (Fig. 2h, i). These results were confirmed by direct visualization of FITC-tagged EN2 uptake (Supplementary Fig. S4). In summary, redox levels modulate the two steps of EN2 trafficking in an uneven manner: low levels of $H_2O_2$ stimulate internalization and reduce secretion, while high levels of $H_2O_2$ have the opposite effects. The dual role of $H_2O_2$ levels on EN2 trafficking observed in cell culture nicely fits with the in vivo effects of Nox-i treatment presented above (Fig. 1h): the nuclear accumulation in producing cells close to the MHB and the reduction of Eng spreading through the tectum are best explained by an inhibition of Eng2 secretion supporting the view that $H_2O_2$ levels directly regulate Eng2 distribution in the zebrafish embryonic tectum.

**Cysteine 175 is involved in the redox regulation of Engrailed spreading**. Reversible oxidation of cysteine residues is the main target of ROS action within proteins, affecting their conformation and activity[29]. Only one cysteine is conserved through evolution in Engrailed proteins. It is located at position 175 in chicken EN2, next to the hexapeptide motif, that is essential both for EN2 transcriptional activity[30] and intercellular transfer[20] (Fig. 3a). To test whether this conserved cysteine is required for EN2 functions, we first evaluated the DNA-binding and transcriptional activities of a C175S EN2 mutant ($EN2_{C>S}$) using as negative control $EN2_{W>K}$, a W169>K,W>172>K double mutant deficient for transcriptional activity and intercellular transfer[20,31]. In the electrophoretic mobility shift assay (EMSA), $EN2_{C>S}$ bound its target nucleic acid sequence in the presence of its PBX co-factor with the same efficiency as EN2 (Fig. 3b). In co-transfection experiments, $EN2_{C>S}$ stimulated transcription from the MAP1b promoter (a known target of Engrailed[32]) to the same extent as EN2 (Fig. 3c). The transfer ability of $EN2_{C>S}$ was then analyzed in secretion and internalization assays. As shown in Fig. 3d–g, $EN2_{C>S}$ transfer between cells was strongly impaired at both the secretion and internalization steps. Importantly, $EN2_{C>S}$ secretion became insensitive to $H_2O_2$ level modulation (Fig. 3e, f). Reducing EN2 cysteine by DTT decreased its internalization to $EN2_{C>S}$ levels, which was insensitive to DTT treatment (Fig. 3h). Part of this effect might involve the ability of EN2 to dimerize as a covalent EN2 dimer (Supplementary Fig. S5a). Indeed, promoting covalent redox-insensitive EN2 homodimerization using a chemical cross-linker (Supplementary Fig. S5b) stimulated internalization compared to native EN2 (Fig. 3i). In summary, the $EN2_{C>S}$ mutation had no broad effect on the transcriptional activity of EN2, but it

(via D-Ala addition) stimulated EN2 secretion (Fig. 2b), whereas decreasing them by either expressing $CAT_{ΔC}$ or treating the cells with purified CAT or Nox-i reduced EN2 secretion (Fig. 2c–e).

To study EN2 internalization, purified recombinant HiBiT-tagged EN2 was added to the medium, and its specific delivery into the cytosol was quantified by light production resulting

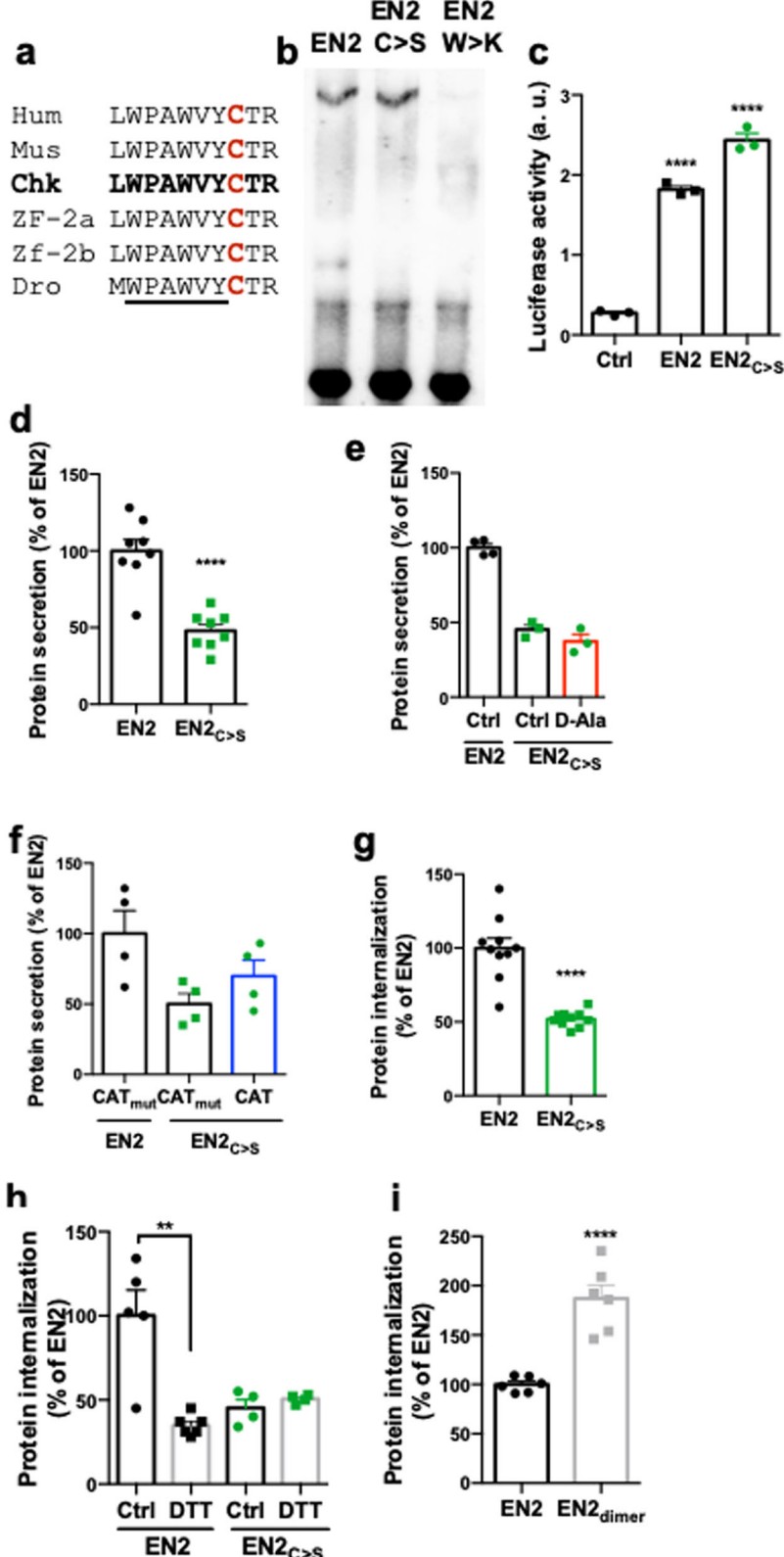

**Fig. 3 A conserved cysteine is involved in the redox regulation of Engrailed transfer. a** Conserved cysteine residue in Engrailed 2 proteins across species (the hexapeptide is underlined). **b** Gel-shift assay comparing the DNA-binding properties of EN2, EN2$_{C>S}$ and DNA-binding deficient EN2$_{W>K}$. **c** Quantification of EN2$_{C>S}$ and EN2 transcriptional activity on the MAP1b promoter. **d**–**f** Quantification of EN2 or EN2$_{C>S}$ secretion via the transRUSH method from control cells **d** or cells expressing Lck-DAO with or without D-Ala **e** or inactive or active Catalase (CAT$_{mut}$ or CAT, respectively) **f**. Quantification of EN2 or EN2$_{C>S}$ internalization **g** after DTT pretreatment **h** or EN2 dimerization **i**. *$p$-value $\leq$ 0.05; **$p$-value $\leq$ 0.01; ***$p$-value $\leq$ 0.001; and ****$p$-value $\leq$ 0.0001.

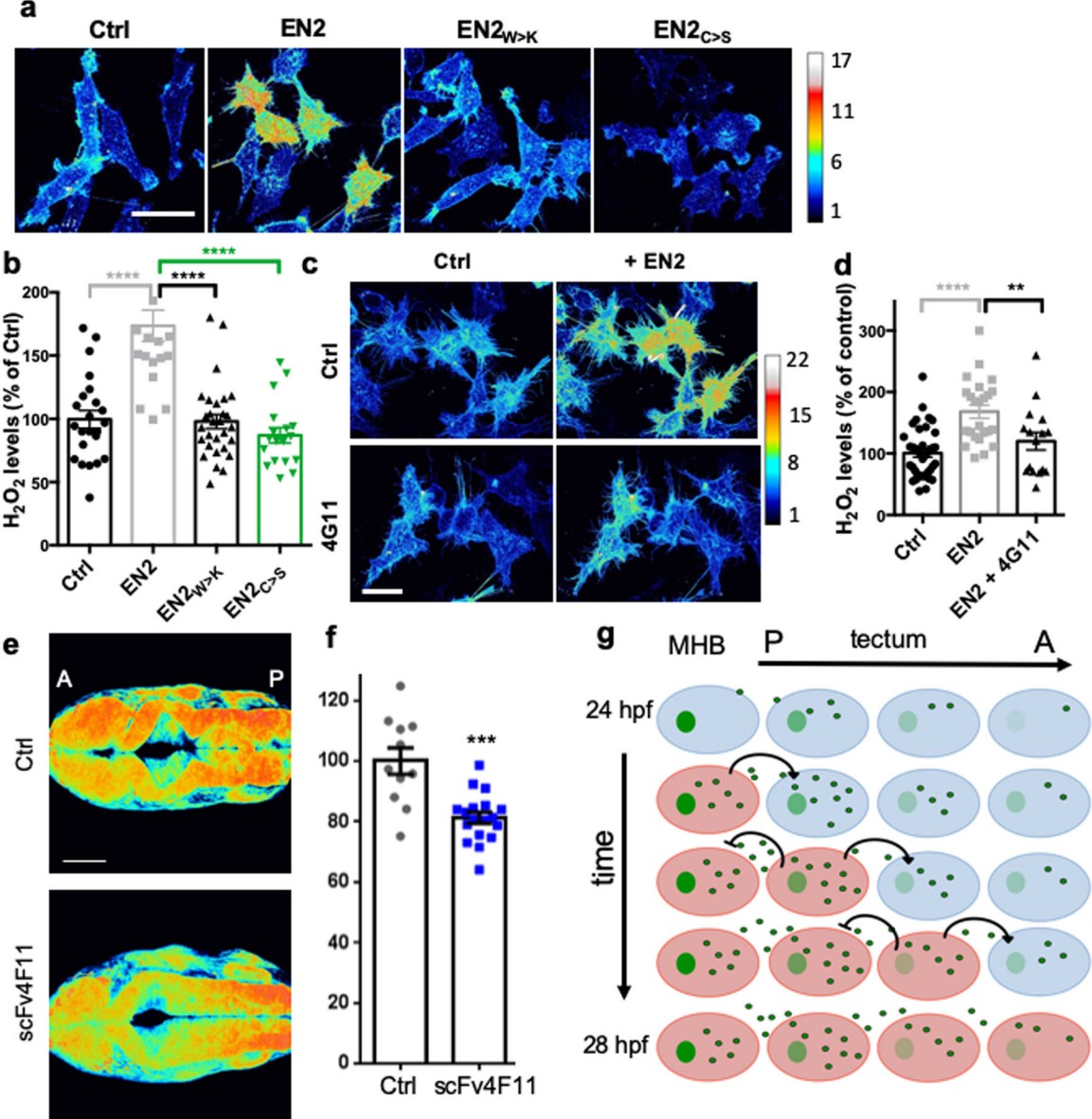

**Fig. 4 Engrailed 2 internalization increases H$_2$O$_2$ levels. a, b** H$_2$O$_2$ imaging and quantification in cells treated with extracellular EN2, EN2$_{W>K}$, or EN2$_{C>S}$. H$_2$O$_2$ levels were inferred as in Fig. 1. **c, d** H$_2$O$_2$ imaging and quantification of cells treated with extracellular EN2 and the 4G11 antibody directed against Engrailed. **e, f** Representative images **e** and quantification **f** of H$_2$O$_2$ levels (inferred as in Fig. 1) in the tecta of zebrafish embryos (26 hpf) expressing scFv4F11 or mCherry as a control. **g** Synthetic model of H$_2$O$_2$–Engrailed interaction during tectum development (H$_2$O$_2$: red; Engrailed: green). In the tectum, H$_2$O$_2$ levels increase over time and are graduxly distributed from the MHB to the anterior part of the embryo (A: anterior, P: posterior). Engrailed is released from cells with high H$_2$O$_2$ levels and enters cells with lower H$_2$O$_2$ levels, where it stimulates the production of H$_2$O$_2$. This results in the establishment of a polarized gradient of Engrailed. *$p$-value ≤ 0.05; **$p$-value ≤ 0.01; ***$p$-value ≤ 0.001; and ****$p$-value ≤ 0.0001.

lessened EN2 intercellular transfer and abolished its sensitivity to H$_2$O$_2$.

**Engrailed 2 internalization increases H$_2$O$_2$ levels**. Signaling pathways are not just linear chains of events, but use crosstalk, feedback, and reciprocal interactions to control biological processes[33]. In this context, because Engrailed is known to influence the oxidative metabolism of recipient cells[11], we wondered whether its trafficking[34] could have an impact on intracellular H$_2$O$_2$ levels. To address this question, EN2 was added to the medium of cells expressing HyPer throughout their cytoplasm. EN2 addition, but not of either mutant proteins deficient for

internalization (EN2$_{W>K}$ and EN2$_{C>S}$), induced a quick increase in H$_2$O$_2$ levels (Fig. 4a, b). To verify that EN2 internalization was responsible for this increase, this step was specifically blocked by preincubation with the 4G11 monoclonal antibody[19]. Under these conditions, EN2 addition no longer enhanced intracellular H$_2$O$_2$ levels (Fig. 4c, d). EN2 import is thus necessary and sufficient to modify cytoplasmic H$_2$O$_2$ levels in recipient cells ex vivo. We then asked whether Eng intercellular transfer also modulates H$_2$O$_2$ levels in vivo. Extracellular anti-En single-chain antibody 4F11scFv, which blocks Engrailed transfer[35], was expressed in the Eng domains using eng2a mini-enhancer[36] in zebrafish embryos expressing HyPer7. This led to

a strong reduction in $H_2O_2$ levels in the tecta of embryos expressing the blocking antibody, but not of embryos expressing a fluorescent protein as a control (Fig. 4e, f).

In summary, Engrailed internalization enhanced intracellular levels of $H_2O_2$, modifying the properties of the recipient cells.

## Conclusion

Together, these results demonstrate that Engrailed is transferred from cell to cell in the zebrafish tectum in an $H_2O_2$-dependent manner, and itself acts rapidly to establish an $H_2O_2$ gradient in vivo. We propose that the global wave of $H_2O_2$ that takes place during development (Fig. 1d, e) sets the spatio-temporal window for heightened Engrailed transfer. Engrailed internalization increases $H_2O_2$ levels in recipient cells, making them more competent to secrete Engrailed and reducing Engrailed uptake once they have reached an $H_2O_2$ threshold. Both events propagate the Engrailed signal forward (posterior to anterior), which leads to a directional spread of the Engrailed gradient.

As a consequence, the extent and intensity of the morphogenetic action of Engrailed mediated via its non-cell autonomous diffusion[37] is shaped by its reciprocal interactions with the main redox signaling molecule (Fig. 4g). We still do not know if this finding can be generalized to other HPs endowed with paracrine activity[13], but if so, may give new insights into how tissue morphogenesis and cell metabolism influence each other. Given the role of HPs expression in evolution, it is tempting to consider how such a mechanism may have contributed to link the second rise in atmospheric oxygen to the concomitant increase in metazoan complexity[38].

## Methods

**DNA constructs, recombinant proteins, and biochemical assays**. Details of the DNA constructs are given in Supplementary Table S1. His6-tagged recombinant proteins were produced in BL21 (DE3) grown in MagicMedia (Invitrogen) 24 h, 28 °C and purified on HisTrap columns (GE Healthcare) by Imidazole gradient elution on AKTA Prime according to manufacturer instructions. Following tag removal by incubation with PreScisson protease (6 h, 4 °C), the protein was purified on Heparin column (GE Healthcare), eluted by NaCl gradient and dialyzed for 2 days (20 mM phosphate buffer, 100 mM NaCl, pH 7.5). For protein FITC labeling, 100 μM of dialyzed purified protein was incubated with a two-fold molar excess of fluorescein isothiocyanate in carbonate buffer (50 mM pH 9.5, 100 mM NaCl) overnight at 4 °C and free FITC was removed by dialysis (48 h, 4 °C). The efficacy of FITC incorporation was determined by SDS–PAGE and spectral analysis. The molecular FITC:protein ratio was confined to a range between 1.5 and 2 for all proteins. Gel-shift assays were performed as in ref. [39]. Redox-insensitive En2 covalent dimerization was obtained with the homobifunctional crosslinker 1,8-bis(maleimido)diethylene glycol (Thermo #22336), according to manufacturer instructions.

**Fish care and manipulation**. Fish husbandry: Zebrafish were maintained and staged as previously described[40]. Experiments were performed using the standard AB wild-type strain. The embryos were incubated at 28 °C. Developmental stages were determined and indicated as hours postfertilization (hpf). The animal facility obtained permission from the French Ministry of Agriculture for all the experiments described in this manuscript (agreement no. C 75-05-12).

Nucleic acid injection: Plasmids (2 ng/μl) were injected into the one-cell stage embryos with 20 ng/μl transposase mRNA to induce DNA recombination into the genome. Transgenic embryos were screened for fluorescent protein expression prior to the analysis. mRNA synthesis was performed using the mMESSAGE mMACHINE transcription kit from Thermofisher, Inc. Equivalent volumes of 75 ng/μl mRNA were injected into one-cell stage embryos to induce ubiquitous expression. EN2-ERT2 fusions were activated at 90% epiboly by adding cyclofen as described in ref. [20].

Pharmacological treatments: To decrease $H_2O_2$ levels, embryos were incubated in VAS-2870 (Nox-i) (100 nM) from Enzo Life Sciences (#BML-El395-0010, Enzo Life Sciences, Inc.; Farmingdale, NY, USA) or an equivalent amount of DMSO as a control for the duration of the time-lapse analysis.

Whole-mount immunostaining: Embryos were fixed with paraformaldehyde (4%, 2 h, room temperature) in PBS. The embryos were permeabilized with cold acetone (−20 °C, 10 min), washed in Triton X-100 (0.8% in PBS, room temperature), and saturated with PBS containing 10% sheep serum, 1% DMSO (Sigma), and 0.8% Triton X-100 (1 h, room temperature) before incubation with the primary antibody (4D9 anti-Engrailed antibody was deposited to DSHB by

Goodman C, and used at a 1:500 dilution, overnight, 4 °C) and secondary antibody (Alexa Fluor-488 goat anti-mouse IgG, diluted 1:1500, with DAPI diluted 1:500, overnight, 4 °C).

RNA extraction and quantitative PCR: Total mRNA was extracted form 30 embryos per sample using Monarch total RNA miniprep kit according to the manufacturer's protocol (NEB). 500 ng of total mRNA was reverse-transcribed by superscript II reverse transcriptase using oligo(dT) primers (Invitrogen). Quantitative PCR was performed using LightCycler® 480 detection system from Roche, Taqman gene expression master mix and Taqman probes from Applied Biosystem (rpl13: Dr03119261_m1; en2a: Dr03079901_m1; en2b: Dr03118700_m1). eng2a and eng2b gene expressions were determined using ΔΔCt method and normalized to rpl13 levels. Each sample was tested in triplicate. The run protocols were performed according to manufacturer's recommendations.

Embryo imaging: The larvae were anesthetized in tricaine solution and embedded in low-melting agarose (0.5%). Imaging was performed with a CSU-W1 Yokogawa spinning disk coupled to a Zeiss Axio Observer Z1 inverted microscope equipped with a sCMOS Hamamatsu camera and a ×25 (Zeiss 0.8 Imm DIC WD: 0.57 mm) oil objective. DPSS 100 mW 405 nm and 150 mW 491 nm lasers and a 525/50 bandpass emission filter were used.

Quantification and statistical analyses: Total Engrailed was quantified by measuring the mean fluorescence along an 80 pixel-width line starting from the MHB border on Z-projections based on MHB-crossing slices. Nuclear and extranuclear Engrailed levels were analyzed by using the DAPI staining as a mask. Nuclear over extranuclear ratio was then calculated on the raw fluorescence values. Ordinary one-way ANOVA followed by Tukey's multiple comparison test was performed to evaluate the significant differences between the conditions along the tectum. $H_2O_2$ levels were quantified by measuring the mean ratio value of HyPer7 per time point and normalized by the mean ratio value determined before treatment. A $t$-test was then performed to statistically determine the differences between the conditions over time.

**Cell culture, ex vivo $H_2O_2$ manipulation, and transcriptional activity analysis**. Cell culture experiments were performed with HeLa cells grown in DMEM supplemented with 10% fetal bovine serum. Transient transfection was performed with Lipofectamine 2000 (Life Technologies) according to the manufacturer's instructions. The cells were cultured for an additional 24 h before being processed for analysis. HeLa cells constitutively expressed a tetracycline repressor (HeLa-Flp-In/T-Rex, Life Technologies), and doxycycline was used to induce protein expression when tetracycline-sensitive expression plasmids were transfected. Stable HeLa cells in which protein expression was controlled by doxycycline were prepared using the HeLa Flp-In cell line, which was kindly provided by Stephen Taylor[41]). A list of the stable cell lines used in this study is given in Supplementary Table S3.

Pharmacological treatments: To decrease $H_2O_2$ levels, cells were either treated with extracellular CAT (Sigma-Aldrich #C1345, 4 U/mL) or pretreated for 1 h with VAS-2870 (Nox-i) (10 μM) (#BML-El395-0010, Enzo Life Sciences, Inc.; Farmingdale, NY, USA) or an equivalent amount of DMSO as a control. To increase $H_2O_2$ levels, cells expressing D-amino acid oxidase (DAO) were treated with 10 mM D-alanine (Sigma-Aldrich #A7377) before the internalization or secretion assays were performed. Blocking anti-Engrailed monoclonal antibody 4G11 was deposited to the DSHB by Jessel TM and Brenner-Morton S.

Quantitative secretion assay: Cells (13,000 per well) stably expressing doubly tagged EN2 (SBP-EN2-HiBiT) were plated on 96-well plates (Greiner Bio-one) coated with polyornithine (10 μg/mL) and induced for constitutive protein expression with doxycycline. After 10 h, the cells were transfected with bidirectional expression plasmids of the transRUSH series making use of the sCMV enhancer, all expressing transmembrane fusions of LgBiT (outside) and core streptavidin (inside) downstream the CMV minimal promoter, and expressing in the opposite direction—downstream the sCMV promoter—either one of Lck-tagged DAO, active CAT devoid of a peroxisome signal, or its inactive counterpart. After 24 h, media were removed, and cells were incubated with fresh medium at 37 °C. Secretion was induced with biotin (100 μM final), and luciferase activity was measured 1 h later with a 96-well plate luminometer (Tristar, Berthold) as described in the HiBit assay kit (Promega). The cells were then lysed to measure intracellular protein expression. Normalization with biotin-untreated wells enabled us to calculate the secretion index and report the secretion efficiency.

Qualitative internalization assay: Cells (30,000 per well) were plated on μ-slide six-well plates (Ibidi). After 24 h, the medium was removed, and cells were incubated with a fluorescent protein (1 μM) diluted in DMEM without serum for 30 min at 37 °C before visualization on a CSU-W1 Yokogawa spinning disk coupled to a Zeiss Axio Observer Z1 inverted microscope equipped with a sCMOS Hamamatsu camera with a ×63/1.4 oil WD: 0.17 mm objective. Cells were analyzed following addition of Trypan blue (0.1% final concentration), an efficient quencher of all extracellular fluorescence (and that of permeabilized cells), to visualize intracellular staining.

Quantitative internalization assay: Cells (90,000 per well) stably expressing LgBiT were plated in 24-well culture dishes. After 24 h, the purified protein fused to HiBiT was added to the medium for 30 min at 37 °C before incubating the cells with trypsin and removing them. After centrifugation, cells were resuspended in PBS and 2% FBS, and the luciferase activity of internalized protein was measured

with a 96-well plate luminometer (Tristar, Berthold) with a HiBiT assay kit (Promega).

$H_2O_2$ imaging with the HyPer probe: HyPer fluorescence was excited with 501/16 and 420/40 bandpass excitation filters, and the corresponding YFP emission was measured using a 530/35 bandpass emission filter. Spinning-disk images were acquired using a ×63 objective (×63/1.4 oil WD: 0.17 mm) on a Spinning-Disk CSU-W1 (Yokogawa) equipped sCMOS Hamamatsu 2048×2048 camera. To calculate the HyPer ratio, images were treated as previously described[42].

Transcriptional activity: MAP1B promoter activation by EN2 or EN2C>S was quantitated in co-transfection experiments as previously described[32] except that the reporter construct expressed Nanoluciferase instead of Luciferase.

Statistics and reproducibility: Data were analyzed using GraphPad Prism 6 and expressed as the mean ± standard error of the mean (SEM). Statistical significance was calculated using a two-sided paired Student's t-test. For multiple conditions, ordinary one-way ANOVA followed by Tukey's multiple comparison test was performed to evaluate the significant differences. Sample sizes and number of replicates are given in Supplementary Table 4. For each experiment, at least two independent experiment with similar results were performed.

**Reporting summary**. Further information on research design is available in the Nature Research Reporting Summary linked to this article.

## Data availability
Source data are provided with this paper. The DNA constructs and cell lines and transgenic fish are available upon request. All other data underlying the findings of the study are available from the corresponding author upon reasonable request.

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

## Acknowledgements
We thank Francesca Meda for preliminary experiments in $H_2O_2$ level manipulation in cell culture, and Filippo Del Bene for fruitful comments on the manuscript. We also thank anomymous reviewers for insightful suggestions. This work was funded by CNRS, INSERM, Collège de France, and Université de Paris and grant #075-15-2019-1789 from the Ministry of Science and Higher Education of the Russian Federation.

## Author contributions
S.V. and A.J. conceived the project and designed the experiments. I.Q., M.T., I.A., A.J. and M.V. prepared the D.N.A. constructs used in this study. I.A., M.T. and C.R. performed the experiments. V.B. and V.V.P. provided the HyPer7 sensor and useful advice. I.A., M.T., C.R., M.V., A.J., and S.V. analyzed the experimental data. A.P., M.V., A.J. and S.V. wrote the paper.

## Competing interests

The authors declare no competing interests.
