## [Peer Review File · Communications Biology]

Reviewers' comments:

Reviewer #1 (Remarks to the Author):

Brief Summary:

The authors of this paper investigate the relationship between the levels and spatial dynamics of H₂O₂ and the distribution of Engrailed homeoprotein in the midbrain during patterning in the zebrafish embryo. Their overall hypothesis is that redox signaling via H₂O₂ levels participates in embryonic patterning, possibly through affecting gradients of homeobox proteins. After describing the H₂O₂ and Engrailed protein gradients in wildtype embryos, they show that reduction of H₂O₂ levels by chemical means in embryos leads to what appears to be reduced spread of Engrailed from its point of origin in the MHB. They test this idea by using an established intercellular trafficking model in HeLa cells to demonstrate that high levels of H₂O₂ in cells are associated with increased secretion and reduced uptake of Engrailed protein, while low levels are associated with decreased secretion and increased uptake—which is consistent with reduced spread of Engrailed in H₂O₂ deficient embryos previously mentioned. The authors conclude that redox levels modulate Engrailed trafficking in an asymmetric manner. They then go on to show that a single conserved cysteine within the hexapeptide essential for Engrailed transactivational activity and intercellular transfer is necessary for the H₂O₂ regulation of Engrailed trafficking, but not for its DNA-binding activity. Conversely, the authors also show that treatment of HeLa cells with wildtype, but not cysteine mutant or antibody-blocked Engrailed, results in Engrailed uptake and a concomitant increase of H₂O₂ levels in these cells. Using transgenic zebrafish in which an Engrailed-blocking antibody is expressed, the authors then show that blocking Engrailed transfer leads to a decrease in H₂O₂ levels in the tectum of these embryos. This leads to their conclusion that intercellular transfer of Engrailed modulates H₂O₂ levels *in vivo*. Finally, the authors present a model that attempts to integrate all these findings, in which they propose that a wave of H₂O₂ sets the temporal and spatial stage for heightened Engrailed transfer, which then increases internalization of Engrailed and H₂O₂ levels in neighbor recipient cells, leading to directional spread of the Engrailed gradient. At the same time, negative feedback occurs on secreting cells once their recipient neighbors accumulate enough H₂O₂ to reduce uptake and increase their own secretion.

Overall Impression:

This is a focused study that draws several important conclusions about the connection between H₂O₂ levels and Engrailed distribution in the developing midbrain. The conclusions are for the most part well-supported by quality data generated both *in vivo* and *in vitro* and using several methods of approach. The written text could be more explicit in parts and draw further connections between different sets of data. Overall, the study is well-conducted and is suggestive of a broader involvement of redox signaling in embryonic patterning, making it a significant report to the field of interest. I recommend acceptance with minor revisions.

Major comments/recommendations: It would be easier for the reader to grasp the major significant observations and conclusions of this report if the authors add more explicit and synthetic statements that lay out the trends seen in the Results and describe more fully the interpretations of data and the interconnections between the different sets of data in the Discussion. For example:

1) Describe the wildtype gradients of Engrailed proteins in Results for Fig 1, explicitly state the direction/polarity of both the nuclear and the cytoplasmic gradients along the A-P axis and over time from 24 hpf to 28 hpf, and integrate this information with the total concentration changes of the protein over time and space. Do you see any extracellular staining of Engrailed anywhere or anytime? (if so, this might tie into your model in the end). Do the same for the H₂O₂ gradient—state explicitly the direction/polarity of the gradient in space and time. Use a summary statement to tie in all these gradients with one another: from A-P, there is increased total Eng, increased nuc Eng, and increased H₂O₂; from 24 hpf to 28 hpf there is reduced total Eng and increased H₂O₂. You can then refer to this summary when you explain the final model. It seems consistent with these observations, except for the decrease in total Eng over time—your model doesn't show that. Does cyto-nuc distribution differ from 24 hpf to 28 hpf in WT embryos? Your model seems to imply this, but

is there evidence for it? Finally, In Nox-i 26 hpf embryos, is there any change in cytoϕnuc distribution of Eng?

2) In the Results text summarizing Fig 2: these results suggest we might expect an increase in extracellular EN with high H₂O₂ in embryos? If so, do you detect this in Fig 1? Results with low H₂O₂ support what you see in vivo in Nox-i embryos, so I would explicitly state this.

3) The explanation of Fig 4 in the text of Results needs to make explicit connection among the different sets of data presented so far. Describe all gradients in both spatial and temporal dimensions, and lay out the events in the order you imagine them happening. Global wave of H₂O₂ (increasing AϕP and 24-28 hpf) sets stage for increased transfer of Eng to neighboring recipient cells, which then increase their own internal H₂O₂ levels, propagating the Eng signal forward (PϕA); this provides polarity to the Eng gradient. Conversely, there is negative feedback, as recipient cells increase internal H₂O₂ and decrease their own uptake and increase their own secretion. What is not clear from your model, is how it fits in with your total Eng protein levels going down over time (Fig 1), which is not shown in the model. Also, unclear why you show what you do about nuc localization in the model, especially in the upper righthand corner.

Minor comments/recommendations: I suggest some additions to the Figure legends and Methods. For instance:

4) Fig 1: indicate MHB in image 1a & 1c; state how Eng & H₂O₂ levels were normalized in 1b & 1c; mention whether embryos are 26 hpf in 1d.

5) Fig 2: mention that you are using Hela cells for this set of experiments.

6) Fig 4: label explicitly both axes of your model: time from 24 hpfϕ28hpf; space from AϕP(MHB?)

7) Methods: It is unclear how exactly you measured total Eng amounts—be more explicit in how this was done.

Reviewer #2 (Remarks to the Author):

This paper addresses the unknown question about whether hydrogen peroxide is a cellular signaling agent in neural development. Using zebrafish as a model, the authors were able to show that in fact this molecular does in fact regulate the distribution of a homeoprotein through the modulation of its intracellular transfer. They go further to show a cysteine residue that is involved in that process. Their writing is clear, the graphs and images are excellent, and their conclusions strong yet not overreaching.

Reviewer #3 (Remarks to the Author):

In this novel and important paper, the authors report that the homeoprotein Engrailed is transferred from cell to cell in the zebrafish tectum in an H₂O₂ dependent manner, and itself acts rapidly to establish an H₂O₂ gradient in vivo. Specifically, Engrailed is released from cells with high H₂O₂ and enters cells with lower H₂O₂, where it stimulates the production of H₂O₂. This results in the establishment of a polarized gradient. The authors identify a conserved cysteine on Engrailed that regulates Engrailed transcriptional activity and transfer between cells. The work is primarily in vivo, and should be of considerable interest within the developmental morphogenesis and redox signaling communities.

1. The abstract should include 'endogenous H₂O₂' so someone quickly reading through it does not assume H₂O₂ is being added exogenously to the zebrafish larvae.

2. For those not as familiar with zebrafish, a diagram of the relationship of the anatomical structures shown in Fig. 1a and Fig. 1c would be helpful.

3. Fig. 1a shows Eng2 in nuclei at the posterior side of the tectum at 24h. Presumably, the nuclear localization is important to the function of Engrailed. Does this nuclear/cytoplasmic distribution change at 26 and 28hrs as the H₂O₂ levels increase?

4. Fig. 1b The gradient of Engrailed expression charted in b is not visually apparent in Fig. 1a. I recommend better explaining how the images were quantified to give the data shown in 1b and/or including an image where the Engrailed gradient is clearly visible. This is an important aspect of the model in Fig. 4, so should be shown more convincingly.

5. Fig. 2, text p.4 line 7: Is it possible to quantify the levels of H₂O₂ present in the D-Ala, CAT, and Nox-I conditions to substantiate this statement?

6. I recommend strengthening the abstract and the concluding paragraph of the paper/model discussion by being more specific. 'Modulates' can mean increases or decreases, and it would be helpful to the reader to clarify the direction of the modulation. What I think is happening: Engrailed is secreted from cells with high H₂O₂ and taken up by cells with low H₂O₂, where it leads to increased H₂O₂ production. This spreads the H₂O₂ gradient in a polarized way. (Text p.4 line 7 is fairly clear, but a more pedantic explanation would be helpful).

7. I appreciate that the authors are careful in this paper not to overstate their results or speculate too much beyond the data shown, however, it would be helpful to the reader if an idea of the mechanism by which Engrailed induces an increase in H₂O₂ was included.

Minor:

1. Abstract, line 25: participate in the patterning of the embryo

2. Fig. 2a: Another sentence regarding how the method works would be helpful in the figure legend. 3. Fig. 4: Explain the model a bit more fully in the figure legend.

4. p5, line 12: remove comma after levels

5. p5, line 17-19, too many dashes in this sentence, replace some with commas 6. p6, line 2-3: remove dashes, just cite ref 36

We would like to thank the reviewers for their useful comments and suggestions that helped us to further develop the discussion and to provide new data in order to improve our manuscript.

Reviewer #1 (Remarks to the Author):

Brief Summary:

The authors of this paper investigate the relationship between the levels and spatial dynamics of H₂O₂ and the distribution of Engrailed homeoprotein in the midbrain during patterning in the zebrafish embryo. Their overall hypothesis is that redox signaling via H₂O₂ levels participates in embryonic patterning, possibly through affecting gradients of homeobox proteins. After describing the H₂O₂ and Engrailed protein gradients in wildtype embryos, they show that reduction of H₂O₂ levels by chemical means in embryos leads to what appears to be reduced spread of Engrailed from its point of origin in the MHB. They test this idea by using an established intercellular trafficking model in Hela cells to demonstrate that high levels of H₂O₂ in cells are associated with increased secretion and reduced uptake of Engrailed protein, while low levels are associated with decreased secretion and increased uptake—which is consistent with reduced spread of Engrailed in H₂O₂ deficient embryos previously mentioned. The authors conclude that redox levels modulate Engrailed trafficking in an asymmetric manner. They then go on to show that a single conserved cysteine within the hexapeptide essential for Engrailed transactivational activity and intercellular transfer is necessary for the H₂O₂ regulation of Engrailed trafficking, but not for its DNA-binding activity. Conversely, the authors also show that treatment of Hela cells with wildtype, but not cys-mutant or antibody-blocked Engrailed, results in Engrailed uptake and a concomitant increase of H₂O₂ levels in these cells. Using transgenic zebrafish in which an Engrailed-blocking antibody is expressed, the authors then show that blocking Engrailed transfer leads to a decrease in H₂O₂ levels in the tecta of these embryos. This leads to their conclusion that intercellular transfer of Engrailed modulates H₂O₂ levels in vivo. Finally, the authors present a model that attempts to integrate all these findings, in which they propose that a wave of H₂O₂ sets the temporal and spatial stage for heightened Engrailed transfer, which then increases internalization of Engrailed and H₂O₂ levels in neighbor recipient cells, leading to directional spread of the Engrailed gradient. At the same time, negative feedback occurs on secreting cells once their recipient neighbors accumulate enough H₂O₂ to reduce uptake and increase their own secretion.

Overall

Impression:

This is a focused study that draws several important conclusions about the connection between H₂O₂ levels and Engrailed distribution in the developing midbrain. The conclusions are for the most part well-supported by quality data generated both in vivo and in vitro and using several methods of approach. The written text could be more explicit in parts and draw further connections between different sets of data. Overall, the study is well-conducted and is suggestive of a broader involvement of redox signaling in embryonic patterning, making it a significant report to the field of interest. I recommend acceptance with minor revisions.

Major comments/recommendations: It would be easier for the reader to grasp the major significant observations and conclusions of this report if the authors add more explicit and synthetic statements that lay out the trends seen in the Results and describe more fully the interpretations of data and the interconnections between the different sets of data in the Discussion. For example:

1) Describe the wildtype gradients of Engrailed proteins in Results for Fig 1, explicitly state the direction/polarity of both the nuclear and the cytoplasmic gradients along the A-P axis and over time from 24 hpf to 28 hpf, and integrate this information with the total concentration changes of the protein over time and space.

R1a: We now provide a temporal analysis of Engrailed cellular distribution, which emphasizes the differences between the gradient shapes from 24 to 28 hpf

Do you see any extracellular staining of Engrailed anywhere or anytime? (if so, this might tie into your model in the end).

R1b: We fully agree that directly visualizing the extracellular staining of Engrailed would have been interesting. We tried to perform immunodetection of extracellular Engrailed in embryos without cell permeabilization. To do so, we could not rely on our previous protocols (dissecting imaginal disks or chicken or zebrafish neural tube) because the early brain is excessively fragile and, most importantly, because dissection would alter the redox balance. We were thus limited to try the removal of yolk and to test different fixation conditions. With no success. We verified with fluorescent dextran and the fluorescent derivative of the primary antibody that the main obstacle is the problem of accessibility of the antibody. It would clearly take time to grope for a non-invasive permeabilization method. Instead we quantified nuclear Engrailed versus extranuclear Engrailed using DAPI as a mask. This is an interesting proxy because we know from previous studies that Engrailed nuclear export gives it access to a non-conventional secretion compartment (Maizel et al., 1999, *Development* 126:3183-90; Wizenmann et al. *Neuron* 64: 355-366, 2009)). These results are now presented in new Figure 1c and 1h. The distribution of Engrailed over time (Figure 1c) most probably results from compounded causes (not exclusively H₂O₂, see below answer R2) but Figure 1h clearly shows that a reduction of H₂O₂ levels strongly modifies the nuclear/extranuclear distribution of Engrailed at the MHB. It appears that with low levels of H₂O₂ Engrailed stays in the nucleus at the MHB with limited transfer to the extracellular space and spreading into the tectum.

Do the same for the H₂O₂ gradient—state explicitly the direction/polarity of the gradient in space and time. Use a summary statement to tie in all these gradients with one another: from A¹P, there is increased total Eng, increased nuc Eng, and increased H₂O₂; from 24 hpf to 28 hpf there is reduced total Eng and increased H₂O₂.

R1b': We added a table in Supplementary information (new Table S1) to summarize these variations.

You can then refer to this summary when you explain the final model. It seems consistent with these observations, except for the decrease in total Eng over time—your model doesn't show that.

R1c: We performed quantitative RT-PCR for *eng2a* and *eng2b* at 24 hpf and 28 hpf (new figure S2) and no significant differences were observed for these mRNAs between the two stages. The stability of Engrailed might differ depending on its cellular behavior, being higher in the nucleus than during its intercellular transfer. This is fully compatible with the observed retention of Engrailed in the nucleus in embryos treated with Nox-i (new figure 1h) and with the higher amount of total Engrailed in the same condition (Figure 1g).

Does cyto¹nuc distribution differ from 24hpf to 28 hpf in WT embryos? Your model seems to imply this, but is there evidence for it? Finally, In Nox-i 26 hpf embryos, is there any change in cyto¹nuc distribution of Eng?

R1d: As mentioned above, we could not quantify extracellular Engrailed. However, extranuclear Engrailed quantification shows that a reduction in H₂O₂ levels enhances the proportion of Engrailed in the nucleus (new Figure 1h).

2) In the Results text summarizing Fig 2: these results suggest we might expect an increase in extracellular EN with high H₂O₂ in embryos? If so, do you detect this in Fig 1? Results with low H₂O₂ support what you see *in vivo* in Nox-i embryos, so I would explicitly state this.

R2: Unfortunately, we do not have access to the extracellular form of Engrailed *in vivo* (see above R1). The quantification of extranuclear Engrailed provided in new Figure 1c shows that Eng behaviour changes at 26 hpf, as expected, but the patterns at 28 and 24 hpf are identical, even though the amount of H₂O₂ is raising between 24 and 28 hpf. One possibility is that H₂O₂ is not the only determinant of

the shape of the gradient, and that compensatory mechanisms start being important between 26 hpf and 28 hpf. Another one is that above a given threshold, H₂O₂ has less effect or no effect on Engrailed secretion. We are now performing experiments to test this hypothesis.

3) The explanation of Fig 4 in the text of Results needs to make explicit connection among the different sets of data presented so far. Describe all gradients in both spatial and temporal dimensions, and lay out the events in the order you imagine them happening. Global wave of H₂O₂ (increasing A Δ P and 24-28 hpf) sets stage for increased transfer of Eng to neighboring recipient cells, which then increase their own internal H₂O₂ levels, propagating the Eng signal forward (P Δ A); this provides polarity to the Eng gradient. Conversely, there is negative feedback, as recipient cells increase internal H₂O₂ and decrease their own uptake and increase their own secretion. What is not clear from your model, is how it fits in with your total Eng protein levels going down over time (Fig 1), which is not shown in the model. Also, unclear why you show what you do about nuc localization in the model, especially in the upper righthand corner.

R3: We added intermediate conclusions and highlighted more explicitly time and space in the model (new Figure 4g and new table S1).

Minor comments/recommendations: I suggest some additions to the Figure legends and Methods. For instance:

4) Fig 1: indicate MHB in image 1a & 1c; state how Eng & H₂O₂ levels were normalized in 1b & 1c; mention whether embryos are 26 hpf in 1d).

R4: This information was added to the figure and the legend.

5) Fig 2: mention that you are using Hela cells for this set of experiments.

R5: It was added to the title of the figure legend.

6) Fig 4: label explicitly both axes of your model: time from 24 hpf Δ 28hpf; space from A Δ P(MHB?)

R6: The scheme was modified accordingly.

7) Methods: It is unclear how exactly you measured total Eng amounts—be more explicit in how this was done.

R7: Details were added in Methods section.

Reviewer #2 (Remarks to the Author):

This paper addresses the unknown question about whether hydrogen peroxide is a cellular signaling agent in neural development. Using zebrafish as a model, the authors were able to show that in fact this molecular does in fact regulate the distribution of a homeoprotein through the modulation of its intracellular transfer. They go further to show a cysteine residue that is involved in that process. Their writing is clear, the graphs and images are excellent, and their conclusions strong yet not overreaching.

We thank Reviewer 2 for his/her positive evaluation of the manuscript

Reviewer #3 (Remarks to the Author):

In this novel and important paper, the authors report that the homeoprotein Engrailed is transferred from cell to cell in the zebrafish tectum in an H₂O₂ dependent manner, and itself acts rapidly to establish an H₂O₂ gradient in vivo. Specifically, Engrailed is released from cells with high H₂O₂ and enters cells with lower H₂O₂, where it stimulates the production of H₂O₂. This results in the

establishment of a polarized gradient. The authors identify a conserved cysteine on Engrailed that regulates Engrailed transcriptional activity and transfer between cells. The work is primarily in vivo, and should be of considerable interest within the developmental morphogenesis and redox signaling communities.

We thank Reviewer 3 for his/her laudatory appreciation of the manuscript and have introduced the suggested editorial changes.

1. The abstract should include 'endogenous H₂O₂' so someone quickly reading through it does not assume H₂O₂ is being added exogenously to the zebrafish larvae.

R1: The text was modified accordingly.

2. For those not as familiar with zebrafish, a diagram of the relationship of the anatomical structures shown in Fig. 1a and Fig. 1c would be helpful.

R2: Positions of the MHB and the tectum are now indicated.

3. Fig. 1a shows Eng2 in nuclei at the posterior side of the tectum at 24h. Presumably, the nuclear localization is important to the function of Engrailed. Does this nuclear/cytoplasmic distribution change at 26 and 28hrs as the H₂O₂ levels increase?

R3: Indeed, nuclear localization is mandatory for Engrailed transcriptional and epigenetic activities, while its presence in the cytoplasm is associated to its ability to regulate translation and to its export. We have now analysed the pattern of nuclear/extranuclear distribution, which changes over time and space (new Figure 1c) and is highly altered when we artificially lower H₂O₂ levels (new Figure 1h). A comment has been added in the text.

4. Fig. 1b The gradient of Engrailed expression charted in b is not visually apparent in Fig. 1a. I recommend better explaining how the images were quantified to give the data shown in 1b and/or including an image where the Engrailed gradient is clearly visible. This is an important aspect of the model in Fig. 4, so should be shown more convincingly.

R4: We have now included a picture of Engrailed staining without DAPI in Supplementary Figure S1 with a lookup table that emphasizes quantitative differences. DAPI was indeed somehow obscuring Engrailed signal, and the gradient is much more noticeable here in this new Figure.

5. Fig. 2, text p.4 line 7: Is it possible to quantify the levels of H₂O₂ present in the D-Ala, CAT, and Nox-I conditions to substantiate this statement?

R5: With the tools at hand we could not measure the absolute H₂O₂ levels in these different conditions. But we can quantitate their relative variations and this is stated in the text.

6. I recommend strengthening the abstract and the concluding paragraph of the paper/model discussion by being more specific. 'Modulates' can mean increases or decreases, and it would be helpful to the reader to clarify the direction of the modulation. What I think is happening: Engrailed is secreted from cells with high H₂O₂ and taken up by cells with low H₂O₂, where it leads to increased H₂O₂ production. This spreads the H₂O₂ gradient in a polarized way. (Text p.4 line 7 is fairly clear, but a more pedantic explanation would be helpful).

R6: The text was modified accordingly

7. I appreciate that the authors are careful in this paper not to overstate their results or speculate too

much beyond the data shown, however, it would be helpful to the reader if an idea of the mechanism by which Engrailed induces an increase in H₂O₂ was included.

R7: We have preliminary data suggesting that the PIP2/Rac1 axis is a relay for the modulation of NADPH oxidase by Engrailed, but our data are too preliminary for publication.

Minor:

1. Abstract, line 25: participate in the patterning of the embryo
2. Fig. 2a: Another sentence regarding how the method works would be helpful in the figure legend.
3. Fig. 4: Explain the model a bit more fully in the figure legend.
4. p5, line 12: remove comma after levels
5. p5, line 17-19, too many dashes in this sentence, replace some with commas
6. p6, line 2-3: remove dashes, just cite ref 36

All these suggestions were introduced in the text.

REVIEWERS' COMMENTS:

Reviewer #1 (Remarks to the Author):

The authors have satisfactorily addressed all my questions, concerns, and suggestions. I heartily recommend publication of this manuscript in Communications Biology.

Reviewer #3 (Remarks to the Author):

The authors have addressed all of my previous concerns.
I recommend going over the new content again for clarity and to catch typos and clunky sentences, since I noticed quite a few issues with a quick read-through.

Abstract: I recommend reducing the wordiness of the first two sentences.
Line 1 'role of' should be 'role for' redox signaling

Main text:

p.3 Line 3: For some reason, there is a \$ sign in my PDF (I read the track changes version).

Line 4-9: The sentence beginning 'Ratiometric probes' is a run-on sentence and should be shortened, divided, or otherwise written more clearly.

Line 16: missing a space

p.4 Line 1: 'polarized traffic' not 'polarity traffic'.

p. 5 Line 10: The passive voice in this 'The temporal analysis...' sentence makes it seem like someone else did this work.

p. 5 Line 25 and elsewhere (lots of these): missing spaces between number and unit

p. 7 Line 10 The sentence should start with a capital letter.

p. 9 Line 6 'take place' should be 'takes place'.

p. 9 Line 13 'insights into' not 'insights on'